# Massive Memorization with Hundreds of Trillions of Parameters for Sequential Transducer Generative Recommenders

**Zhimin Chen**[1][†][*]**, Chenyu Zhao**[1][†]**, Ka Chun Mo**[1]**, Yunjiang Jiang**[1]**,
Jane H. Lee**[2][‡]**, Khushhall Chandra Mahajan**[1]**, Ning Jiang**[1]**,
Kai Ren**[1]**, Jinhui Li**[1][*]**, Wen-Yun Yang**[1][*]

[1]Meta, [2]Yale University

## Abstract

Modern large-scale recommendation systems rely heavily on user interaction history sequences to enhance the model performance. The advent of large language models and sequential modeling techniques, particularly transformer-like architectures, has led to significant advancements recently (e.g., HSTU, SIM, and TWIN models). While scaling to ultra-long user histories (10k to 100k items) generally improves model performance, it also creates significant challenges on latency, queries per second (QPS) and GPU cost in industry-scale recommendation systems. Existing models do not adequately address these industrial scalability issues. In this paper, we propose a novel two-stage modeling framework, namely *VIrtual Sequential Target Attention* (VISTA), which decomposes traditional target attention from a candidate item to user history items into two distinct stages: (1) user history summarization into a few hundred tokens; followed by (2) candidate item attention to those tokens. These summarization token embeddings are then cached in storage system and then utilized as sequence features for downstream model training and inference. This novel design for scalability enables VISTA to scale to lifelong user histories (up to one million items) while keeping downstream training and inference costs fixed, which is essential in industry. Our approach achieves significant improvements in offline and online metrics and has been successfully deployed on an industry leading recommendation platform serving billions of users.

## 1 Introduction

Personalized recommendation systems are now integral to digital platforms like streaming services, e-commerce, and social media, where they boost user engagement and drive key metrics such as click-through rates (CTR), session duration, and retention. The success of these systems hinges on their ability to accurately predict user preferences by processing and interpreting vast user histories.

While traditional recommendation models, such as collaborative filtering (Sarwar et al., 2001) and matrix factorization (Koren et al., 2009), laid the groundwork for personalized recommendation, they often struggle to scale and capture long-term user behaviors. Deep learning introduced more powerful solutions, and the recent integration of large language models (LLMs) and sequential modeling techniques such as transformers (Section 2) has marked a significant leap forward, enabling the capture of intricate interactions across vast user histories.

---

[†]These authors contributed equally to this work.
[‡]Work done during internship at Meta.
[*]Corresponding authors: {zhimin, jinhui, wenyun}@meta.com

Figure 1: VISTA replaces standard attention with a two-stage process, allowing downstream models to compute only the highly efficient second stage.

In the domain of recommendation systems, two primary types of sequence modeling techniques have been explored: full user sequence modeling, as seen in Hierarchical Sequential Transduction Units (HSTU) (Zhai et al., 2024), and target-specific sequence sampling, as seen in Search-based Interest Modeling (SIM) (Pi et al., 2020) and its subsequent works (Chang et al., 2023; Si et al., 2024). Both approaches have demonstrated success in enhancing recommendation system performance by harnessing users' extensive historical interactions.

Despite their success, full sequence modeling suffers from the computational cost of scaling. Modeling full user interaction sequences which are usually on the scale of $O(100K)$ in length often leads to enormous computational costs and latency issues, which are very challenging for industrial recommendation systems that need to train $O(10B)$ to $O(100B)$ examples per day and have strict latency upper limits during inference. As a result, the full sequence modeling methods such as HSTU (Zhai et al., 2024) are hindered by high computational costs, limiting its widespread adoption across industries where many companies are still short of GPU capacities.

The second approach, target-specific sequence sampling, has been extensively explored through a series of seminal works, including SIM (Pi et al., 2020), TWIN (Chang et al., 2023), and TWIN V2 (Si et al., 2024). These studies have demonstrated the effectiveness of leveraging user historical interaction sequences. However, subsequent research in this direction has encountered two significant challenges: (1) bridging the gap between attention to the target-specific shortened sequence and the full user sequence, which, however, was partially addressed in follow-up work TWIN (Chang et al., 2023); and (2) the computational cost increases linearly with the number of candidates to predict at inference time, due to the independent target-specific sequences. These two challenges remain largely unresolved, primarily due to the inherent design limitations of SIM-style models.

Addressing the challenges of scalability in recommendation systems will assist with their widespread adoption. In this paper, we propose a novel two-stage modeling framework, *VIrtual Sequential Target Attention* (VISTA), designed to overcome the scalability challenges. The first stage compresses the ultra-long user interaction history into a few hundred of summarization embeddings (see Fig. 1); the second stage serves as efficient candidate aware target attention mechanism that leverages the summarization from the first stage for final prediction. The first stage occurs only during foundational model training, where the resulting summarization embeddings are cached to conceptually represent user embeddings. Consequently, downstream model training and inference only need to perform the second stage: computing attention between a candidate item and these cached embeddings, instead of processing the full user interaction history. This approach significantly reduces the computational complexity for downstream models, especially during inference, at the cost of additional storage. In practice, this is a worthwhile trade-off, as the cost of GPU computation remains multiple orders of magnitude higher than the cost of storage.

As a summary of our contributions, to the best of our knowledge we are the first to propose:

- A two-stage attention framework to decouple foundational model training and downstream model training and inference, which enables us to leverage ultra-long user histories for better recommendation model performance in industrial-scale systems,
- A quasi-linear attention formulation tailored for recommendation models,
- A generative sequential reconstruction loss in recommendation models, and
- A practical embedding delivery system successfully deployed in an industrial-scale platform.

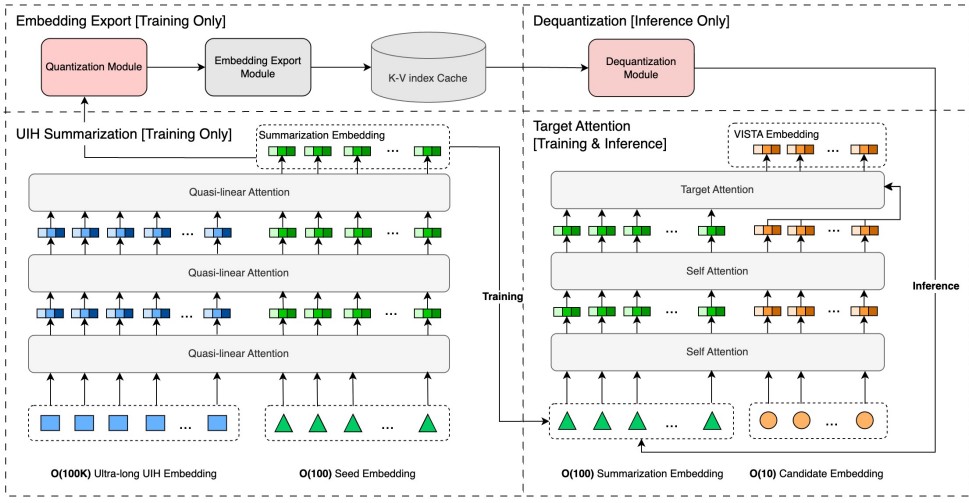

Figure 2: An overview of VISTA architecture.

## 2 RELATED WORK

**Hierarchical Sequential Transduction Unit (HSTU).** A significant advancement in this area is the Hierarchical Sequential Transduction Unit (HSTU) (Zhai et al., 2024), which reframes recommendation as a sequential transduction problem. Designed specifically for high-cardinality, non-stationary streaming recommendation data, HSTU surpasses traditional models in both accuracy and efficiency. This architecture allows recommendation systems to scale to trillions of parameters, leading to substantial gains in predictive performance.

**Transformer Architectures in Recommendation Systems.** The application of transformer architectures in recommendation systems has been explored extensively. By leveraging the self-attention mechanism, transformers can model complex user-item interactions over time, facilitating more nuanced and personalized recommendations (Subbiah and Aggarwal, 2024). The Deep Interest Network (DIN) (Zhou et al., 2018) and its follow-up work, Search-based Interest Modeling (SIM) (Pi et al., 2020; Chang et al., 2023; Si et al., 2024), leverage lifelong sequential behavior data. This approach employs search-based mechanisms, also known as General Search Units (GSUs), to select a small subset of relevant interactions from the user's historical sequence that are pertinent to the target item followed by a standard transformer network, referred to as Exact Search Units (ESUs), to compute the final target item representation. Notably, this method enables the modeling of user behavior data with lengths reaching up to hundreds of thousands (Pi et al., 2020). Other methods (Liu et al., 2023) preprocess user histories into groups and attend to the group embeddings, and separately attend to subsequences in the user history relevant to the target item.

**Linear Complexity Attention Mechanisms.** Apart from Flash Attention (Dao et al., 2022; Dao, 2023) that is designed to improve the efficiency of the softmax attention mechanisms, there is a new trend to explore linear complexity attention mechanisms. Katharopoulos et al. (2020) first proposes linear attention. By applying matrix multiplication associative property, it enables a change in computation order from $(QK^T)V$ to $Q(K^TV)$, reducing computation complexity from $O(N^2)$ to $O(N)$ with respect to sequence length $N$. Recently, Lightning Attention v1 (Qin et al., 2024a) and v2 (Qin et al., 2024b) propose a light network which contains Gated Linear Attention (GLA) and Simple Gated Linear Unit (SGLU) to make linear attention more practical. Another branch of linear complexity work, namely state space model (SSM), has been widely studied. Mamba (Gu and Dao, 2024) is a pioneering work in SSM and widely used in many real-world applications, followed by Hydra (Hwang et al., 2024) which is the double-headed version of Mamba to address non-causal scenarios.

## 3 METHOD

Here we introduce the details of VISTA's two cascaded modules: ultra-long user interaction history (UIH) sequence summarization and target-aware attention, followed by details of a practical linear complexity self-attention and generative sequence reconstruction loss. We then explain how VISTA's design enables the scaling, storage, and processing of industry-scale user history sequences through its embedding delivery system.

### 3.1 MODEL ARCHITECTURE OVERVIEW

As illustrated in Figure 2, the VISTA architecture employs distinct workflows for training and inference. During training, the computationally expensive UIH summarization module runs to generate summary embeddings. These embeddings are then quantized and exported to a large key-value cache in $O(100)$ terabytes to $O(1)$ petabytes. For inference, this expensive step is bypassed entirely. Instead, the pre-computed embeddings are simply retrieved from the cache and dequantized with minimal distortion. The final component, the target attention module, operates in both phases, using the summarization embeddings and candidate item features to make predictions.

### 3.2 ULTRA-LONG UIH SEQUENCE SUMMARIZATION

In the first stage, we utilize self-attention with virtual seed embeddings to summarize ultra-long UIH sequences. These virtual seeds are initialized randomly as shared parameters across users, which are updated with the model through its interaction with the UIH sequence in the summarization module. The output of the summarization module can be interpreted as user embeddings, encoding individual personalized preferences to inform recommendations. Figure 3 visualizes these summarization embeddings, projected onto the first 2 principal components by principal component analysis (PCA). We can clearly see the separation for users of different countries, with US and Canada overlapping, which is expected.

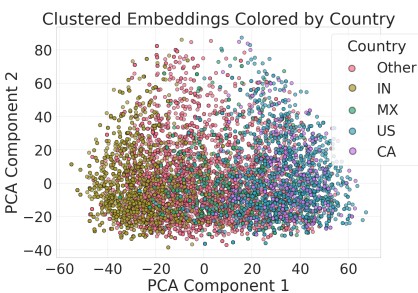

Figure 3: Visualization of UIH summarization embeddings.

However, typical softmax attentions suffer from $O(N^2)$ time complexity, which is prohibitive when dealing with ultra-long sequences ($N > 10k$). Therefore, we propose quasi-linear attention (QLA), a linear time complexity $O(N)$ self-attention mechanism to overcome this issue.

#### 3.2.1 LINEAR ATTENTION WITH CANDIDATE ITEMS FOR RECOMMENDATION

With the emergence of Large Language Models (LLMs), researchers have proposed some linear complexity attention algorithms to accelerate transformer blocks (Katharopoulos et al., 2020; Qin et al., 2024a;b; Han et al., 2024). However in recommendation systems, unlike the text sequences in LLM, a strict rule is that *the candidates cannot attend each other*, since it introduces label leakage due to the fact that the logged candidates typically only form a small subset of the input candidates during inference. Therefore, we propose a linear-complexity self attention mechanism that avoids attention among candidates.

The typical softmax self attention for a UIH sequence $S$ can be formulated as follows

$$\text{SoftmaxAttn}(\text{S} \Rightarrow_{\text{full}} \text{S}) = \text{RowSoftmax}(QK^\top)V$$

where $Q$, $K$ and $V$ have shape $(L, d)$ and $L$ is the sequence length. Then the original linear attention (Katharopoulos et al., 2020) for a UIH sequence $S$ can be written similarly as follows

$$\text{LinAttn}(\text{S} \Rightarrow_{\text{full}} \text{S}) = \text{RowNormalize}(QK^\top)V \tag{1}$$
$$= Q(K^\top V)/\text{RowSum}(QK^\top) = Q(K^\top V)/(Q \, \text{ColSum}(K)^\top).$$

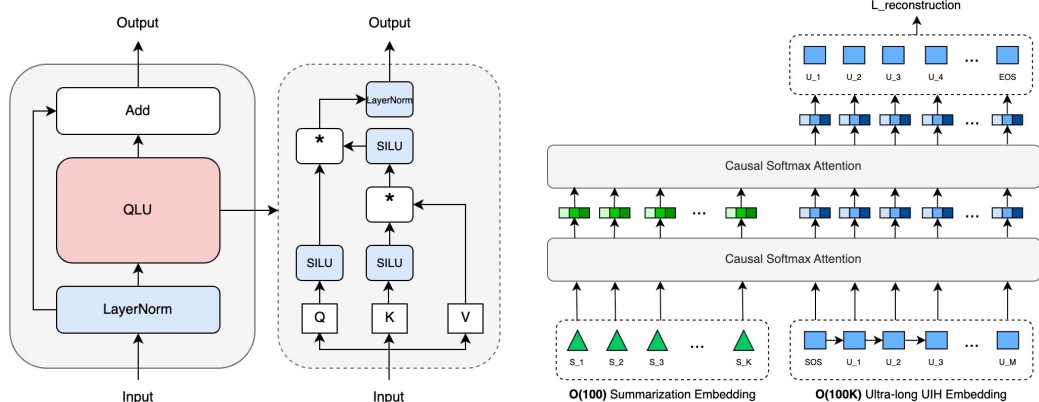

Figure 4: The QLU module.    Figure 5: Generative reconstruction loss.

Note that division / here stands for broadcast division along the rows. The above can be applied to full (bi-directional) self-attention.

In recommendation models, we also have target (candidate) items, let's denote them by $T$. Then we want to compute target attention of $T$ against $K$ and $V$.

$$\text{LinAttn}(T \Rightarrow_{\text{full}} S) = T(K^\top V)/(T\,\text{ColSum}(K)^\top). \tag{2}$$

Note that candidates cannot attend to each other. This is a strict rule in recommendation systems otherwise the model training will fail due to the leakage between candidate items. It gets slightly trickier if we also want each candidate to attend to itself. Instead of $TT^\top T$, the contribution due to the self attention of each target item to itself is given by

$$\text{LinAttn}(T \Rightarrow_{\text{individual}} T) = \text{Diag}(TT^\top)T. \tag{3}$$

### 3.2.2 Quasi-linear Attention for Recommendation

Despite its efficiency, some previous works (Han et al., 2024; 2023) prove that linear attention suffers from insufficient expressive power, making it impractical for real applications. In this section, we introduce quasi-linear attention (QLA) as an empirically effective linear attention algorithm for recommendation. This quasi-linear attention introduces more non-linear complexity in attention computation, addressing the issue of expressive power.

The quasi-linear attention contains two parts: Quasi Linear Unit (QLU) module and Simple Gated Linear Unit (SGLU) module. The QLU module aims to model the interaction of $Q, K, V$ matrices with SiLU non-linear activation as shown in Figure 4. For the SGLU module, we use the same gated function as TransNormerLLM (Qin et al., 2024a).

Accordingly, we need to slightly modify the above linear attention formulation to accommodate this QLU module. For the self attention part we let the user history items attend to one another. Similar as in HSTU (Zhai et al., 2024), SASRec (Kang and McAuley, 2018), and Pinnerformer (Pancha et al., 2022), usually the causal self-attention approach via a triangular mask is used. In our case, we did not find significant difference between causal and full self attention, since the user history items merely serve as features for the final candidate prediction task – their temporal causality is not a strict requirement. Let $\varphi$ denote a non-linear activation function (we use SiLU in our experiments), then the full self quasi-linear attention modified from Eq. (1) is as follows.

$$O[S] = \varphi(Q[S])\varphi(\varphi(K[S])^\top V[S])$$

where $[S]$ denotes the source (user history) portion of the sequence. Note that we remove the RowNormalize operation, similarly as in Lightning Attention (Qin et al., 2024a;b).

For the target portion of the query sequence embeddings, we can similarly apply $\varphi$-linear attention between $Q[T]$ and $K[S], V[S]$. However to be consistent with the self-attention

semantics, we also include an extra term that captures attention to the target item itself. Thus, the final formula for the target portion of the quasi-linear attention, modified from Eq. (2) and (3) is given by

$$O[T] = \varphi(Q[T])\varphi(\varphi(K[S])^\top V[S]) + \Delta(\varphi(Q[T]), \varphi(K[T]))V[T].$$

Here $\Delta(X,Y)_{ij} := \sum_k X_{ik}Y_{ik}\delta_{ij}$ stands for putting the row-wise dot product between the two matrices $X$ and $Y$ of shape $n \times m$ on the diagonal of a square matrix of shape $n \times n$. In order to implement the quasi-linear attention efficiently using the Triton language (Tillet et al., 2019) for optimized GPU computation performance, we also calculate the gradient of the final loss function with respect to the input tensors $Q[S], Q[T], K[S], K[T], V[S], V[T]$, in terms of the gradient with respect to the output tensor $O[S], O[T]$ in Appendix B.

### 3.2.3 Generative Sequence Reconstruction Loss

To further enhance the memorization effects, we also introduce a reconstruction loss (see Fig. 5) to encourage the sequence summarization to fully reproduce the UIH sequence, which we find particularly useful to improve VISTA's performance. Intuitively, to reconstruct the $i$-th UIH item embedding, we are using all the seed embeddings and the UIH item embeddings up to the $(i-1)$-th position. A natural way to accomplish this is via the decoder network, such as the causal transformer decoder, without the softmax layer. Formally,

$$(t_1, \ldots, t_k, v_1, \ldots, v_M) = \mathrm{Decoder}(s_1, \ldots, s_k, u_1, \ldots, u_M).$$

where $s_1, \ldots, s_k$ are the personalized seed embeddings, and $u_1, \ldots, u_M$ are the UIH item embeddings. We can feed their concatenation through the causal softmax attention block (or any other transformer block) to get the output embeddings concatenated as $t_1, \ldots, t_k$ and $v_1, \ldots, v_M$ where $k$ is the number of seeds and $M$ the length of the user history sequence. Then we can simply form the off-by-one mean square error of the $v_i$'s with the $u_i$'s as the construction loss as $L_{\mathrm{reconstruct}} = \sum_{i=1}^{M-1} \|v_i - u_{i+1}\|_2^2$.

Since causal transformer block ensures that the output embedding $v_i$ only depends on $u_1, \ldots, u_i$, there is no leak of information from $u_{i+1}$ to $v_i$. This forces the personalized seed embeddings $s_i$ to maximize information retained of the user history sequence $u_1, \ldots, u_M$. Similar ideas have roots in the Variational Auto-Encoder (Kingma and Welling, 2022), and have appeared in the context of transformer networks recently (Henderson and Fehr, 2022). However to the best of our knowledge, there has not been any explicit use in recommendation. For more discussion on this reconstruction loss, see Appendix C.

### 3.3 Target-aware Attention

As shown in Figure 2, any attention network can technically be used for the target-aware attention stage. Because this step is computationally inexpensive compared to sequence summarization, we selected a standard $O(N^2)$ transformer block, which delivers excellent performance on the compact summary sequences.

## 4 Embedding Delivery System

We emphasize that the VISTA framework is not merely a theoretical model, but a novel industrial model system co-design to support large scale user interaction history sequence learning that can be deployed into the real industry infrastructure with reasonable cost.

Figure 6 outlines the system's end-to-end architecture, which comprises three main stages: (1) online training of the source model using training data stream, (2) delivery of sequence summarization embeddings to downstream models via two routes: a real-time message queue, e.g., Kafka (Kreps et al., 2011) and persistent storage, e.g., Hive (Thusoo et al., 2009), and (3) serving embeddings through a geographically replicated in-memory key-value store. In our system, we update the summarization embeddings on a 2-hour cadence, which was shown to have similar performance compared to using the summarization module directly in online A/B tests. This design ensures both real-time performance and scalability for industrial applications. For scalability, we deliberately compress the user interaction history sequence to $O(100)$ terabytes level, making it feasible to deploy to existing systems.

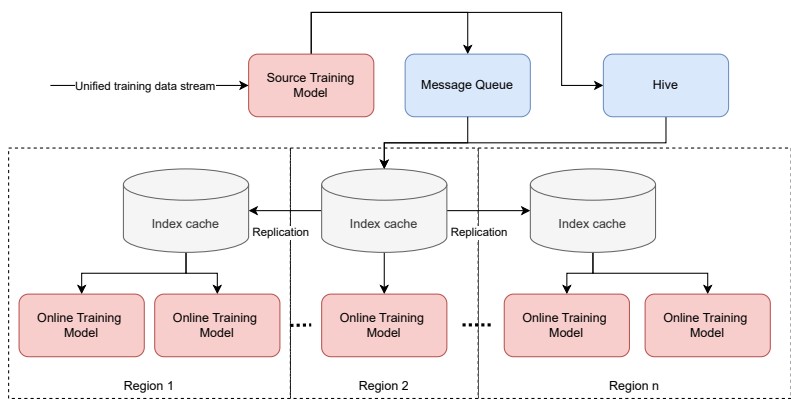

Figure 6: An overview of VISTA sequence summarization embedding delivery system.

## 5 EXPERIMENTS

### 5.1 DATASETS AND EXPERIMENTAL SETUP

The proposed VISTA framework is designed for a large scale real-world dataset, where one needs to train hundreds of billions of examples per day and each user has a history which contains hundreds of thousands of items. While existing public datasets are usually much smaller, we compare our method against several baselines on public datasets in addition to reporting results on real production data.

#### 5.1.1 PUBLIC DATASET AND INDUSTRIAL-SCALE DATASET

We first compare the effectiveness of VISTA against several baseline models on public datasets Amazon-Electronics [1] and KuaiRand-1K [2]. To focus mainly on the effectiveness of the attention mechanism, we compare VISTA against baselines in replacing the attention layers in a common model architecture. All models are implemented, trained, and evaluated under the FuxiCTR [3] framework, focusing on click-through rate prediction. Additionally, we introduce a Minimal Production dataset from real production data, compatible with FuxiCTR having minimal features but with longer sequences up to 2,000.

For industrial-scale offline experimentation, we construct full training and evaluation samples from real production data, with several metrics for engagement, which we denote by "C-Task", "E1-Task", etc. We use 3-day data as the training set and the next 1-day data as the evaluation set in our offline experiment. The scale of training examples per day is at $O(10)$ billion level. The average and maximum UIH sequence lengths are 7,000 and 16,000, respectively. Note that we deploy the model with 12,000 UIH sequence length in online experiments, but we only use 2,000 in offline experiments due to GPU resource constraints.

Table 1: Dataset Statistics

| Dataset | Mean Seq. | Max Seq. |
|---|---|---|
| Amazon-Electronics | 8.93 | 429 |
| KuaiRand-1K | 225.20 | 256 |
| Simplified Prod | 1528.18 | 2,000 |
| Industrial-Scale Data | 7,000 | 16,000 |

---

[1]https://github.com/reczoo/Datasets/tree/main/Amazon/AmazonElectronics_x1
[2]https://kuairand.com/
[3]https://github.com/reczoo/FuxiCTR

Table 2: Comparisons on public and Minimal Production datasets. VISTA-w/-QLA and VISTA-w/o-QLA are the VISTA model with and without quasi-linear attention, respectively.[4]

| Models | Amazon | | KuaiRand | | Minimal Production | |
|---|---|---|---|---|---|---|
| | AUC ($\uparrow$) | NE ($\downarrow$) | AUC ($\uparrow$) | NE ($\downarrow$) | AUC ($\uparrow$) | NE ($\downarrow$) |
| DIN | $0.873 \pm 8e^-4$ | $0.656 \pm 1e^-4$ | $0.744 \pm 0.003$ | $0.864 \pm 0.005$ | $0.632 \pm 0.02$ | $1.048 \pm 0.033$ |
| TTSN | $0.877 \pm 0.005$ | $0.644 \pm 0.010$ | $0.740 \pm 0.003$ | $0.869 \pm 0.004$ | $0.648 \pm 0.005$ | $1.139 \pm 0.156$ |
| MHA | $0.881 \pm 1e^-4$ | $0.634 \pm 0.002$ | $0.743 \pm 0.001$ | $0.863 \pm 0.005$ | $0.630 \pm 0.018$ | $1.049 \pm 0.041$ |
| SASRec | $0.884 \pm 4e^-4$ | $0.627 \pm 0.001$ | $0.742 \pm 0.003$ | $0.868 \pm 0.007$ | $0.605 \pm 0.020$ | $1.129 \pm 0.134$ |
| HSTU | $0.884 \pm 0.001$ | $0.628 \pm 0.001$ | $0.743 \pm 0.001$ | $0.863 \pm 1e^-5$ | $\mathbf{0.668 \pm 0.011}$ | $1.099 \pm 0.048$ |
| VISTA-w/o-QLA | $\mathbf{0.886 \pm 0.002}$ | $\mathbf{0.621 \pm 0.005}$ | $\mathbf{0.744 \pm 0.001}$ | $\mathbf{0.863 \pm 0.003}$ | $0.627 \pm 0.016$ | $\mathbf{1.038 \pm 0.05}$ |
| VISTA-w/-QLA | $0.884 \pm 0.005$ | $0.623 \pm 0.003$ | $0.743 \pm 4e^-4$ | $0.864 \pm 0.001$ | $0.632 \pm 0.013$ | $1.062 \pm 0.076$ |

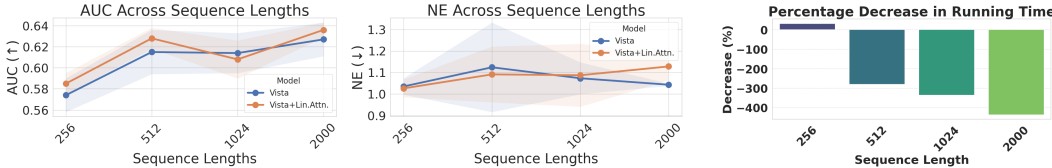

Figure 7: Ablation study on quasi-linear attention by varying sequence length.

### 5.1.2 Baselines and Evaluation Metrics

All models share a common feature embedding layer and MLP block, with consistent hyperparameters, e.g., embedding dimensions, layers, attention heads, for fair comparison. We briefly describe them: (1) Deep Interest Network (DIN) (Zhou et al., 2018) uses attention to adaptively weigh user historical behaviors, (2) Two-Tower Sparse Network (TTSN) (Covington et al., 2016) separately encodes user and item features via two towers, (3) the standard Multi-Head Attention (MHA) (Vaswani et al., 2017), (4) SASRec (Kang and McAuley, 2018) is self-attentive sequential recommendation model that uses the Transformer architecture, and (5) Hierarchical Sequential Transduction Units (HSTU) (Zhai et al., 2024) is an industry proposed transformer-like model designed to capture multi-scale sequential patterns in user behavior sequence.

We use normalized entropy (NE) (He et al., 2014) as our evaluation metric, which calculates the cross entropy between the predicted probabilities and the labels, then normalizes it by the entropy of the constant predictor at label average. We also report the area under curve (AUC) for the traditional setting. Additional details about this section are in Appendix D.

### 5.2 Offline Experimental Results

### 5.2.1 Public Dataset Results

In Table 2, we summarize the comparative results between VISTA and the baseline models. For the Amazon-Electronics dataset, VISTA outperforms the other baselines with the use of quasi-linear attention being the next best model. On the KuaiRand dataset, VISTA slightly outperforms the other models with similar NE to HSTU and MHA. This may suggest that even at smaller sequence lengths, the virtual seeding embeddings slightly help the model performance. On much longer sequences in Minimal Production, we see that HSTU and VISTA perform best demonstrating that both are designed for handling longer sequences.

Figure 7 shows the ablation study results of quasi-linear attention by varying the sequence length on the Minimal Production dataset. We can clearly see that the QLA mechanism significantly reduces the the time to train and evaluate 1 epoch of data, while there are small differences in AUC and NE.

Figure 8 shows the scaling law of increasing number of seed embeddings. We can clearly see that the model performance improves with larger number of seed embeddings, which,

---

[4]Comparisons to the next best model are significant at the $p < 0.001$ level when using a paired t-test. ANOVA tests with multiple next-best models are significant at the $p < 0.001$ level.

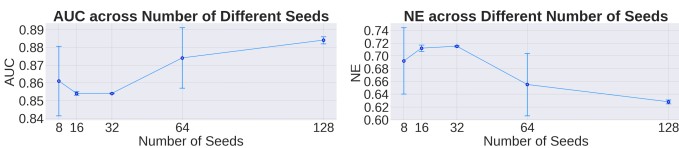
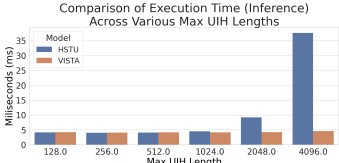

Figure 8: Ablating VISTA across number of seed embeddings on Amazon-Electronics.

Figure 9: Inference time gains with increasing UIH lengths.

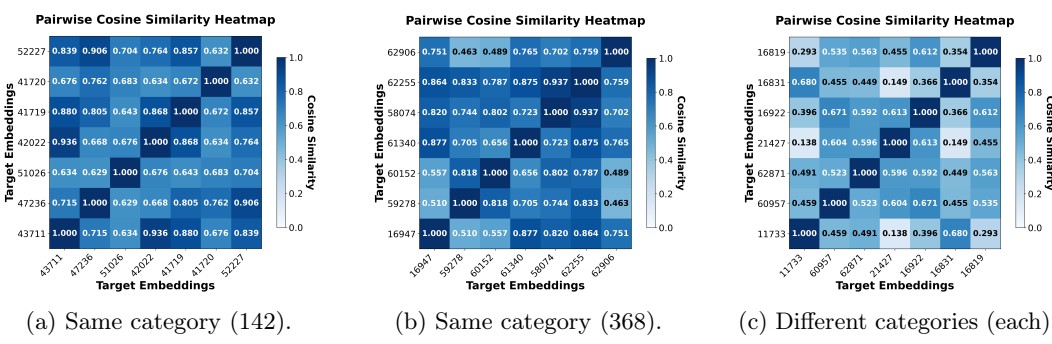

(a) Same category (142).     (b) Same category (368).     (c) Different categories (each).

Figure 10: Pairwise cosine similarity of the output of two-stage attention of VISTA. For the same user, we compare the target attention output for items of similar or different categories.

however, will incur more storage capacity cost in real world production scenario. Thus, in practice it is a tradeoff between model performance and financial cost.

Figure 10 compares the target embeddings, the output of VISTA's two-stage attention modules, for different items given the same user history from Amazon-Electronics. Embeddings between items from the same category are more similar than those from different ones, as expected.

### 5.2.2 Industrial-Scale Dataset Results

In Table 3, we compare our proposed VISTA model with the baseline production model using HSTU as the backbone in both offline and online experiments, on the industrial-scale dataset. In this setting, there are multiple tasks which measure different aspects of engagement information. We report the main consumption task ("C-Task"), and other engagement events ("E1-Task", "E2-Task", and "E3-Task"). To further understand the effectiveness of the model, we also conduct ablation studies by varying the embedding dimension, the number of seeds, and the use of generative reconstruction loss. As shorthands, (1) VISTA stands for the optimized proposed model co-trained with the baseline HSTU model, with 3-layer self-attention, 3-layer target-aware attention, 128 seeds, 256 embedding dimension and 2,000 UIH sequence length. (2) VISTA-128D stands for VISTA model with 128 dimension embedding. (3) VISTA-64Seed stands for VISTA model with 64 seeds. (4) VISTA-w/o-Recon stands for VISTA model without generative reconstruction loss. The results demonstrate that our optimized VISTA configuration (128 seeds, 256 embedding dimension, and 2,000 UIH sequence) significantly outperforms the standalone HSTU baseline for training and evaluation NE metrics.

Table 4 summarizes the performance improvement of QLA compared with the standard self-attention on production dataset, where we can see that the QLA is able to scale up with more layers and longer sequence for better NE metrics and even higher QPS.

Figure 9 shows the VISTA's advantage on inference performance, especially for much longer sequence lengths. This is expected since VISTA's main strength is to cache the UIH summarization. Thus, the most computationally expensive module, UIH summarization module, is deactivated during inference.

Table 3: Offline comparative results with the baseline model and ablation models.

| Models | Training NE (↓) | | | | Eval NE (↓) | | | |
|---|---|---|---|---|---|---|---|---|
| | C-Task | E1-Task | E2-Task | E3-Task | C-Task | E1-Task | E2-Task | E3-Task |
| HSTU | - | - | - | - | - | - | - | - |
| VISTA | **-0.47%** | **-0.82%** | -2.30% | **-1.72%** | **-0.40%** | -1.19% | -2.98% | **-2.23%** |
| VISTA-128D | -0.32% | -0.50% | -1.86% | -1.43% | -0.29% | -1.07% | -2.51% | -1.82% |
| VISTA-64Seed | -0.36% | -0.68% | -1.70% | -1.45% | -0.37% | -1.11% | **-3.01%** | -2.09% |
| VISTA-w/o-Recon | -0.42% | -0.72% | **-2.32%** | -1.69% | -0.29% | **-1.29%** | -3.00% | -2.21% |

Table 4: Comparing VISTA with and without quasi-linear attention.

| Model Variant | Max Seq. | Layers | QPS (↑) | Training NE (↓) | Eval NE (↓) |
|---|---|---|---|---|---|
| VISTA-w/o-QLA | 6,000 | 3 | - | - | - |
| VISTA-w/-QLA | 16,000 | 5 | +5% | -0.1% | -0.13% |

## 5.3 Online A/B Experimental Results

We conducted an online A/B test on our production video recommendation system, using 5% of the entire site traffic during a period of 15 days. The baseline is the HSTU model and we compare with adding the VISTA module, which is the same as our offline experiment setup. Online metrics for the main consumption task "C-Task" and other online onboarding metrics, "O1-Task" and "O2-Task", were significantly improved by 0.5%, 0.2%, 0.04%, respectively. VISTA demonstrated a 94% reduction in inference GPU resource (measured by inference QPS) usage by caching and serving embeddings, rather than re-computing them for every new user request. With a 0.01% "O2-Task" gain considered a substantial improvement on our platform, the VISTA model made realized contributions to the recommendation system.

## 6 Conclusion and Discussions

In this paper, we have proposed the VIrtual Sequential Target Attention (VISTA) framework, a novel two-stage approach that compresses ultra-long user interaction histories into a set of compact embeddings. This design strikes a crucial balance between computational efficiency and predictive accuracy, addressing the latency and scalability challenges of processing ultra-long user sequence data in production systems. VISTA's practical applicability is underscored by its resilience to slight de-synchronization between its stages and its ability to approximate complex transformer architectures without their substantial computational cost. Our empirical evaluations demonstrate that VISTA not only captures the core information within user interactions but also achieves significant improvements across platform metrics. Our plans for future research involve further optimizing VISTA's compression techniques and exploring its applications across other domains to enhance its generalizability.

## Acknowledgement

This work results from a large cross organization collaboration. It would not be possible without contributions from the collaborators and supports from the leaderships as follows (alphabetic order): Zheng-Yong Ang, David Bauer, Connor Chen, Shouwei Chen, Siqiao Chen, Huihui Cheng, Ek Kheng Chung, Litao Deng, Shilin Ding, Chenhao Feng, Kevin Goulding, Liang Guo, Mengyue Hang, Maxwell Lin-He, Xiaoxin He, Chufeng Hu, Jizhou Huang, Yanzun Huang, Han Jiang, Justin Khim, Emma Lin, Zihan Li, Yang Liu, Yining Liu, Li Lu, Wenhan Lyu, Jing Ma, Matt Ma, Jing Qian, Rui Qiao, Chuyu Qiu, Yongxiong Ren, Xinyue Shen, Daisy Shi, Hongzheng Shi, Ge Song, Yisong Song, Wanting Tan, Hao Wan, Meihong Wang, Yanhong Wu, Hong Yan, Yihang Yang, Chuanwei Yi, Christina You, Haoli Zhang, Rui Zhang, Yue Zhang, John Zheng, Xinye Zheng, Lizhen Zhu, Maggie Zhuang.

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

## A  Usage of LLMs Disclosure

In this section, we disclose the usage of LLMs in the preparation of this manuscript. LLMs were used for 1) polishing writing or shortening limited blocks of text and 2) for generating template code for plotting or minor changes of existing code. LLMs were NOT used for retrieval and discovery (e.g., finding related work), research ideation, or any other purpose not explicitly outlined in the above.

## B  Mixed Full Linear Attention

To simplify triton implementation, especially for the gradient computation, our quasi-Linear Attention drops the normalization (RowNormalize) in the usual linear attention, similar to lightning attention. Instead we can mimic what SiLU attention does, by introducing a $1/N$ factor.

$$O = (QK^T) \odot MV/N,$$

where $\odot$ is the Hadamard product (componentwise multiplication of two matrices, and $M = \begin{pmatrix} \mathbf{1}_{n \times n} & \mathbf{0}_{n \times m} \\ \mathbf{1}_{m \times n} & I_m \end{pmatrix}$. To compute this in triton, first break into two parts.

$$Q = \begin{pmatrix} Q[S] \\ Q[T] \end{pmatrix}, \qquad Q[S] \in \mathbb{R}^{n \times d}, \quad Q[T] \in \mathbb{R}^{m \times d},$$

$$K = \begin{pmatrix} K[S] \\ K[T] \end{pmatrix}, \qquad K[S] \in \mathbb{R}^{n \times d}, \quad K[T] \in \mathbb{R}^{m \times d},$$

$$V = \begin{pmatrix} V[S] \\ V[T] \end{pmatrix}, \qquad V[S] \in \mathbb{R}^{n \times d}, \quad V[T] \in \mathbb{R}^{m \times d}.$$

We will divide $n + m$ into $A$ blocks of size $n'$, and divide $n$ into $B$ blocks of size $n''$, so that $Q_i$ are submatrices of dimension $n' \times d$, and $K_j, V_j$ are submatrices of dimension $n'' \times d$.

First we compute

$$(QK[S]^T V[S])_i = Q_i \sum_{j=1}^{B} K[S]_j^\top V[S]_j.$$

Next we compute the target part: we divide $m$ into $C$ blocks of size $m'$ each. For the $j$-th block, it's given by

$$((Q[T]K[T]^\top \odot I_m)V[T])_j = \text{diag}((Q[T]_j \odot K[T]_j)\mathbf{1}_{m' \times 1})V[T]_j.$$

We usually merge the source and target embedding sequences in an interleaved fashion. To avoid HBM/SRAM sync, we probably should keep track of the offsets of the boundary between source and target, and let $n' = m'$, so that for the target part, we will overlap the two computation and obtain

$$O[S]_\ell = Q[S]_\ell \sum_{j=1}^{B} K[S]_j^\top V[S]_j$$

$$O[T]_\ell = Q[T]_\ell \sum_{j=1}^{B} K[S]_j^\top V[S]_j + \text{diag}((Q[T]_\ell \odot K[T]_\ell)\mathbf{1}_{m' \times 1})V[T]_\ell$$

In terms of triton implementation, we will use positive offsets for target, and negative offsets for source, all starting from the boundary offset.

Note that the sum $\sum_{j=1}^{B} K[S]_j^\top V[S]_j$ can be computed first, then multiplied with $Q[S]_\ell$, $Q[T]_\ell$ etc. By choosing the block size $n' = m'$ sufficiently small, and if necessary, also

break the block $V[S]$, $V[T]$ along the columns into smaller dimension $d'|d$, we can ensure all $O[S]_\ell, O[T]_\ell$ blocks can be computed entirely in SRAM with a single for loop.

To replace linear attention with (traditional) SiLU attention for target to source, we need to replace the second line above with

$$O[T]_\ell = \sum_{j=1}^{B} \mathrm{SiLU}(Q[T]_\ell K[S]_j^T) V[S]_j + \mathrm{SiLU}(Q[T]_\ell \odot K[T]_\ell 1_{m' \times 1}) V[T]_\ell.$$

Here we cannot compute all the $O[T]_\ell$ blocks easily, but instead need to have $m/m'$ SM's to compute them separately, otherwise each SM would incur a big for loop of $Bm/m'$ iterations. Given H100 has about 132 SMs and batch size per rank is 512, using more SMs will likely slow things down.

### B.1 GRADIENT COMPUTATION

$$\frac{\partial L}{\partial V} = \mathrm{tr}\left(\left(KQ^\top \frac{\partial L}{\partial O}\right) \odot M^\top\right) / N$$

$$\frac{\partial L}{\partial Q} = \mathrm{tr}\left(\left(KQ^\top \frac{\partial L}{\partial O}\right) \odot M^\top\right)$$

Given that $L = L(O[S], O[T])$, and $O[S]$ and $O[T]$ are disjoint, we can compute

$$dL = \sum_{ij} \frac{\partial L}{\partial O[S]}_{ij} dO[S]_{ij} + \sum_{ij} \frac{\partial L}{\partial O[T]}_{ij} dO[T]_{ij}$$

#### B.1.1 GRADIENT OF V

If we differentiate against $V$, we have

$$dO[S] = Q[S]K[S]^\top dV[S]$$
$$dO[T] = Q[T]K[S]^\top dV[S] + \mathrm{diag}((Q[T] \odot K[T])\mathbf{1}_{T \times 1})dV[T]$$

So,

$$dL = \mathrm{tr}\left(\frac{\partial L}{\partial O[S]}^\top dO[S]\right) + \mathrm{tr}\left(\frac{\partial L}{\partial O[T]}^\top dO[T]\right)$$

$$= \mathrm{tr}\left(\frac{\partial L}{\partial O[S]}^\top Q[S]K[S]^\top dV[S]\right) + \mathrm{tr}\left(\frac{\partial L}{\partial O[T]}^\top (Q[T]K[S]^\top dV[S] + \mathrm{diag}((Q[T] \odot K[T])\mathbf{1}_{T \times 1})dV[T]))\right)$$

$$= \mathrm{tr}\left((\frac{\partial L}{\partial O})^\top QK[S]^\top dV[S]\right) + \mathrm{tr}\left((\frac{\partial L}{\partial O[T]})^\top \mathrm{diag}((Q[T] \odot K[T])\mathbf{1}_{T \times 1})dV[T])\right).$$

So we have that

$$\frac{dL}{dV[S]} = K[S]Q^\top \frac{\partial L}{\partial O}$$

$$\frac{dL}{dV[T]} = \mathrm{diag}((Q[T] \odot K[T])\mathbf{1}_{T \times 1})\left(\frac{\partial L}{\partial O[T]}\right)$$

which means ith row of $\frac{\partial L}{\partial O[T]}$ will be multiplied by ith element of $(Q[T] \odot K[T])\mathbf{1}_{T \times 1}$.

#### B.1.2 GRADIENT OF Q

Next we differentiate against Q,

$$dO[S] = dQ[S]K[S]^\top V[S]$$
$$dO[T] = dQ[T]K[S]^\top V[S] + \mathrm{diag}((dQ[T] \odot K[T])\mathbf{1}_{T \times 1})V[T]$$

Which results in

$$dL = \mathrm{tr}\left((\frac{\partial L}{\partial O[S]})^\top dQ[S]K[S]^\top V[S]\right) + \mathrm{tr}\left((\frac{\partial L}{\partial O[T]})^\top (dQ[T]K[S]^\top V[S]\right)$$
$$+ \mathrm{diag}((dQ[T] \odot K[T])\mathbf{1}_{T\times 1})V[T])).$$

So that

$$\frac{dL}{dQ[S]} = \frac{\partial L}{\partial O[S]}V[S]^\top K[S].$$

To derive $\frac{dL}{dQ[T]}$, we need to pull $dQ[T]$ out of the unconventional expression $\mathrm{diag}((dQ[T] \odot K[T])\mathbf{1}_{T\times 1})$, within the trace operator. Let's first write it in terms of Einstein summation, abbreviation $\frac{\partial L}{\partial O[T]}$, $Q[T]$, $K[T]$, $V[T]$ by $X, Q, K, V$ respectively.

$$\mathrm{tr}(X^\top \mathrm{diag}((dQ \odot K)\mathbf{1}_{T\times 1})V) = \sum_{ijk\ell} X_{ji}dQ_{jk}K_{jk}\delta_{j\ell}V_{\ell i},$$

where $\delta$ is the Kronecker delta matrix given by

$$\delta_{j\ell} = \begin{cases} 1 & \text{if } j = \ell, \\ 0 & \text{otherwise.} \end{cases}.$$

Note that

$$\sum_{i\ell} X_{ji}K_{jk}\delta_{j\ell}V_{\ell i} = \sum_i X_{ji}V_{ji}K_{jk} = (\mathrm{diag}((X \odot V)\mathbf{1}_{T\times 1})K)_{jk}.$$

Thus the second half of the expression for $dL$ (with respect to $dQ[T]$) is given by

$$\mathrm{tr}((K[S]^\top V[S](\frac{\partial L}{\partial O[T]})^\top + (\mathrm{diag}((\frac{\partial L}{\partial O[T]} \odot V[T])\mathbf{1}_{T\times 1})K[T])^\top)dQ[T]).$$

Thus since diagonal matrix is invariant under transposition,

$$\frac{\partial L}{\partial Q[T]} = \frac{\partial L}{\partial O[T]}V[S]^\top K[S] + \mathrm{diag}((\frac{\partial L}{\partial O[T]} \odot V[T])\mathbf{1}_{T\times 1})K[T].$$

### B.1.3 Gradient of K

Similar computation shows

$$\frac{\partial L}{\partial K[S]} = V[S]((\frac{\partial L}{\partial O[S]})^\top Q[S] + (\frac{\partial L}{\partial O[T]})^\top Q[T]) = V[S](\frac{\partial L}{\partial O})^\top Q$$
$$\frac{\partial L}{\partial K[T]} = \mathrm{diag}((\frac{\partial L}{\partial O[T]} \odot V[T])\mathbf{1}_{T\times 1})Q[T]$$

### B.1.4 Summary of Forward Pass and All Gradients

We introduce the notation that produces a diagonal matrix dimension $T \times T$ from two matrices $X, Y$ of dimension $T \times d$:

$$\Delta(X, Y) := \mathrm{diag}((X \odot Y)\mathbf{1}_{T\times 1}) = \{\sum_\ell X_{i\ell}Y_{i\ell}\delta_{ij}\}_{ij}. \tag{4}$$

Then for forward, we have

$$O[S] = Q[S]K[S]^\top V[S] =: Q[S]Z[S]^\top \tag{5}$$
$$O[T] = Q[T]K[S]^\top V[S] + \Delta(Q[T], K[T])V[T] =: Q[S]Z[S]^\top + U[T]V[T] \tag{6}$$

For backward, we have

$$\frac{\partial L}{\partial Q[S]} = \frac{\partial L}{\partial O[S]} V[S]^\top K[S] =: dO[S]Z[S] \tag{7}$$

$$\frac{\partial L}{\partial Q[T]} = \frac{\partial L}{\partial O[T]} V[S]^\top K[S] + \Delta(\frac{\partial L}{\partial O[T]}, V[T])K[T] =: dO[T]Z[S] + X[T]K[T] \tag{8}$$

$$\frac{\partial L}{\partial K[S]} = V[S](\frac{\partial L}{\partial O})^\top Q =: V[S]W^T \tag{9}$$

$$\frac{\partial L}{\partial K[T]} = \Delta(\frac{\partial L}{\partial O[T]}, V[T])Q[T] =: X[T]Q[T] \tag{10}$$

$$\frac{dL}{dV[S]} = K[S]Q^\top \frac{\partial L}{\partial O} =: K[S]W \tag{11}$$

$$\frac{dL}{dV[T]} = \Delta(Q[T], K[T])\frac{\partial L}{\partial O[T]} =: U[T]dO[T]. \tag{12}$$

From equation 7 and equation 8 we see that $Z[S] := V[S]^\top K[S]$ should be an intermediate step to compute, of dimension $d \times d$.

From equation 9 and equation 11, we should compute $W := Q^\top \frac{\partial L}{\partial O}$ first to obtain a $d \times d$ matrix, before pre-multiplying by $K[S]$ or post-multiplying by $V[S]^\top$ and then transpose (or transposing first, then pre-multiplying by $V[S]$).

From equation 6 and equation 12, we see that forward pass should save the intermediate step $Y[T] := \Delta(Q[T], K[T])$. In fact we can just save the diagonal elements, which consumes less memory.

From equation 8 and equation 10, we can also save the intermediate step $X[T] := \Delta(\frac{\partial L}{\partial O[T]}, V[T])$.

While $X[T], Y[T]$ only involve $2Td$ FLOPs, $W$ and $Z$ involve $2Td^2$ FLOPs.

### B.1.5 WITH SHIFTED ELU ACTIVATION

Shifted elu (and its derivative) are defined by

$$\varphi(x) = \begin{cases} x, & \text{if } x \geq 1, \\ e^{x-1}, & \text{if } x < 1, \end{cases} \qquad \text{and} \qquad \varphi'(x) = \begin{cases} 1, & \text{if } x \geq 1, \\ e^{x-1}, & \text{if } x < 1. \end{cases}$$

Given,

$$O[S] = \varphi(Q[S])\varphi(K[S])^\top V[S]$$
$$O[T] = \varphi(Q[T])\varphi(K[S])^\top V[S] + \Delta(\varphi(Q[T]), \varphi(K[T]))V[T]$$

Gradients are given by (partly by guessing via dimension match)

$$\frac{\partial L}{\partial Q[S]} = (\frac{\partial L}{\partial O[S]} V[S]^\top \varphi(K[S])) \odot \varphi'(Q[S])$$

$$\frac{\partial L}{\partial Q[T]} = (\frac{\partial L}{\partial O[T]} V[S]^\top \varphi(K[S])) \odot \varphi'(Q[T]) + \Delta(\frac{\partial L}{\partial O[T]}, V[T])(\varphi'(Q[T]) \odot \varphi(K[T]))$$

$$\frac{\partial L}{\partial K[S]} = (V[S](\frac{\partial L}{\partial O})^\top \varphi(Q)) \odot \varphi'(K[S])$$

$$\frac{\partial L}{\partial K[T]} = \Delta(\frac{\partial L}{\partial O[T]}, V[T])(\varphi(Q[T]) \odot \varphi'(K[T]))$$

$$\frac{dL}{dV[S]} = \varphi(K[S])\varphi(Q)^\top \frac{\partial L}{\partial O}$$

$$\frac{dL}{dV[T]} = \Delta(\varphi(Q[T]), \varphi(K[T]))\frac{\partial L}{\partial O[T]}.$$

## B.2 Activation for Quasi-Linear Attention

The choice of activation function $\varphi$ in Section 3.2.2 is non-linear but otherwise arbitrary and depends on the specific application at hand. The same activation also need not always be used, for instance one can have two different activations $\varphi_1$ and $\varphi_2$ and apply them as:

$$O[S] = \varphi_1(Q[S])\varphi_2(\varphi_1(K[S])^\top V[S]),$$
$$O[T] = \varphi_1(Q[T])\varphi_2(\varphi_1(K[S])^\top V[S]) + \Delta(\varphi_1(Q[T]), \varphi_1(K[T]))V[T].$$

## C More on Reconstruction Loss

The reconstruction loss is a measure of how much information of the full UIH is captured by VISTA's virtual embeddings. We used L2 norm to measure how much we can reconstruct the original UIH given the virtual embedding as input, and we have verified that it is an informative metric for us to quantitatively measure the reconstruction quality.

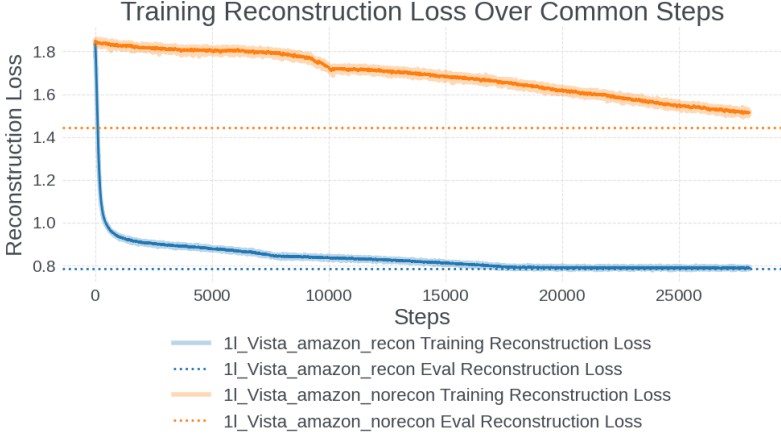

Figure 11: Comparing the reconstruction loss over training steps with and without explicitly including it in the total loss on the Amazon dataset. The dotted line is the final evaluation reconstruction loss.

On the Amazon dataset, for example, we can see that training the VISTA model without explicitly minimizing the reconstruction loss still reduces the reconstruction loss of the learned embedding against the full UIH as the model improves. However, the reconstruction loss plateaus and the model takes longer to converge after some training steps (as our training pipeline supports early stopping). With the explicit introduction of the reconstruction loss, we see a dramatic decrease in the reconstruction loss in the first training steps and faster model convergence. The test metrics also improved by 0.22% AUC and 1.11% NE with the use of the reconstruction loss.

## D Additional Experiment Details

### D.1 Datasets

We include more details about the datasets used in our experiments. The statistics of the sequence features of each dataset are summarized in Table 5.

**Amazon-Electronics.** The Amazon Products and Reviews dataset Hou et al. (2024) contains user reviews, item metadata, and user-item interactions. A subset of this data was preprocessed to make the Amazon-Electronics dataset, which is restricted to electronics items, initiated by Zhou et al. (2018). The data format is relatively simple, with the columns: label, user id, item id, category id, item history, and category history.

Table 5: Dataset Statistics

| Dataset | Mean Seq. | Max Seq. |
|---|---|---|
| Amazon-Electronics | 8.93 | 429 |
| KuaiRand-1K | 225.20 | 256 |
| Simplified Prod | 1528.18 | 2,000 |
| Industrial-Scale Data | 7,000 | 16,000 |

**KuaiRand-1K.** The KuaiRand dataset by Gao et al. (2022) is a sequential recommendation dataset collected from the recommendation logs of the video-sharing mobile app Kuaishou. The KuaiRand-1K subset contains a random sample 1,000 users after removing irrelevant videos. There are 4 million videos remaining in this subset. Our experiments use a subset of all features available in KuaiRand-1K, namely user id, video id, video id history, click history, like history, and lvv (long video view) history.

**Simplified Production and Industrial-Scale Data.** The full production data is too large to be able to run simple experiments quickly (and requires re-implementing baseline models on internal systems). We construct a minimal version of our production data to focus on the sequential recommendation task (e.g., keeping the user interaction history largely intact but removing other features). After preprocessing, this dataset has a mean sequence length of around 1528 and maximum truncated to 2,000.

### D.2 FuxiCTR Framework

We utilize the FuxiCTR library developed by Zhu et al. (2022; 2021) for our traditional sequential setting experiments. As mentioned in the main text, we designed the traditional sequential setting experiments mainly to compare the effectiveness of the attention layers and keep constant other model architecture and hyperparameter choices. (See Figure 12.)

We also report the common hyperparameters used in all the experiment results in Table 13.

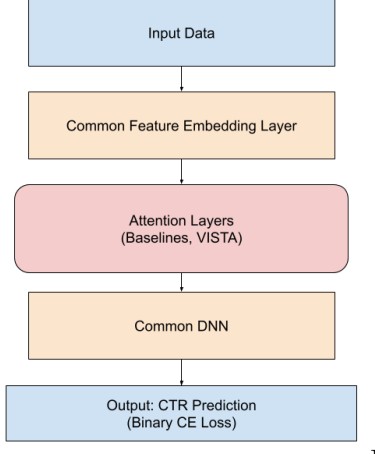

| Hyperparameter | Amazon | KuaiRand | Simplified Prod |
|---|---|---|---|
| Learning Rate | $5.0e{-}4$ | $1.0e{-}4$ | $1.0e{-}3$ |
| Optimizer | Adam | Adam | Adam |
| Batch Size | 1024 | 1024 | 128 |
| Batch Norm | No | No | Yes |
| Early Stop Patience | 4 | 5 | 1 |
| Embedding Regularizer | 0.005 | None | None |
| Embedding Dimension | 64 | 32 | 32 |
| Embedding Initializer | $1e{-}4$ | $1e{-}4$ | $1e{-}4$ |
| MLP Hidden Units | [1024, 512, 256] | [512, 128, 64] | [512, 128, 64] |
| MLP Activations | RELU | RELU | RELU |
| # Attention Heads | 4 | 4 | 4 |
| # Attention Layers | 1 | 1 | 2 |

Figure 12: FuxiCTR Setup.

Figure 13: Common hyperparameters used for traditional setting experiments.

The model-specific parameters for VISTA are the number of seeds and weight for the reconstruction loss, which were set at 128 and 1.0, respectively, for all experiments. No specific hyperparameter tuning was done, mainly relying on using common parameters for all models and repeating across 3 seeds for each model and dataset.

### D.3 More on VISTA's Two-Stage Attention

**Case Study 1: Same User, Different Candidates.** In the following Figure 14, we show the input to the virtual attention layer, the output of the virtual attention layer, and then

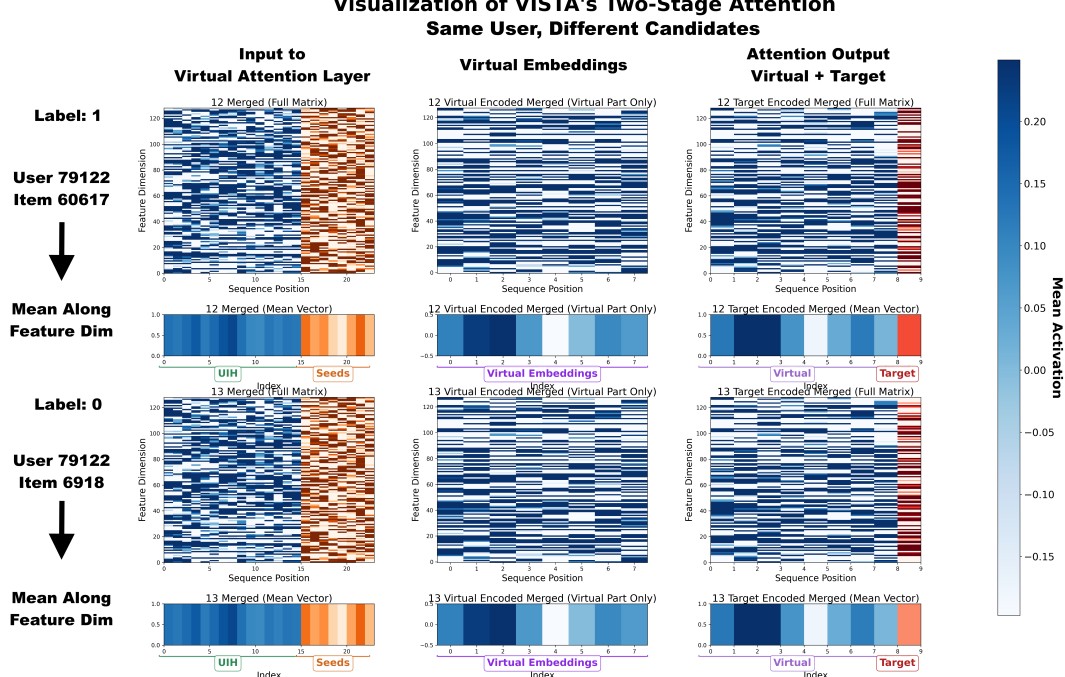

Figure 14: Visualizing the virtual attention and target attention layers for the same user on two different candidates (one positive at the top and one negative at the bottom).

the output of the target attention layer for the same user for two different candidates (one positive at the top and one negative at the bottom) from the Amazon-Electronics dataset. We also reduce these along the feature dimension for compact visualization. Note that since we are looking at two candidates for the same user, the input to the virtual attention layer and the output of the virtual attention layer are identical; these only depend on the user's individual UIH and the virtual seed embeddings which are common between the two. The difference in this case comes at the target attention part. Here we see that the target attention differentiates between positive and negative candidates for this user as evidenced by the differing mean activation for the target embedding.

**Case Study 2: Different UIH Lengths.** We also look at a case study comparing two different users with different UIH histories, one with a very short history (length 4) and another with a slightly longer history (length 12) in Figure 15. Even with very little historical data (UIH length 4 at the top), the virtual seed embeddings appear to help influence the virtual embeddings, which in turn help with model performance.

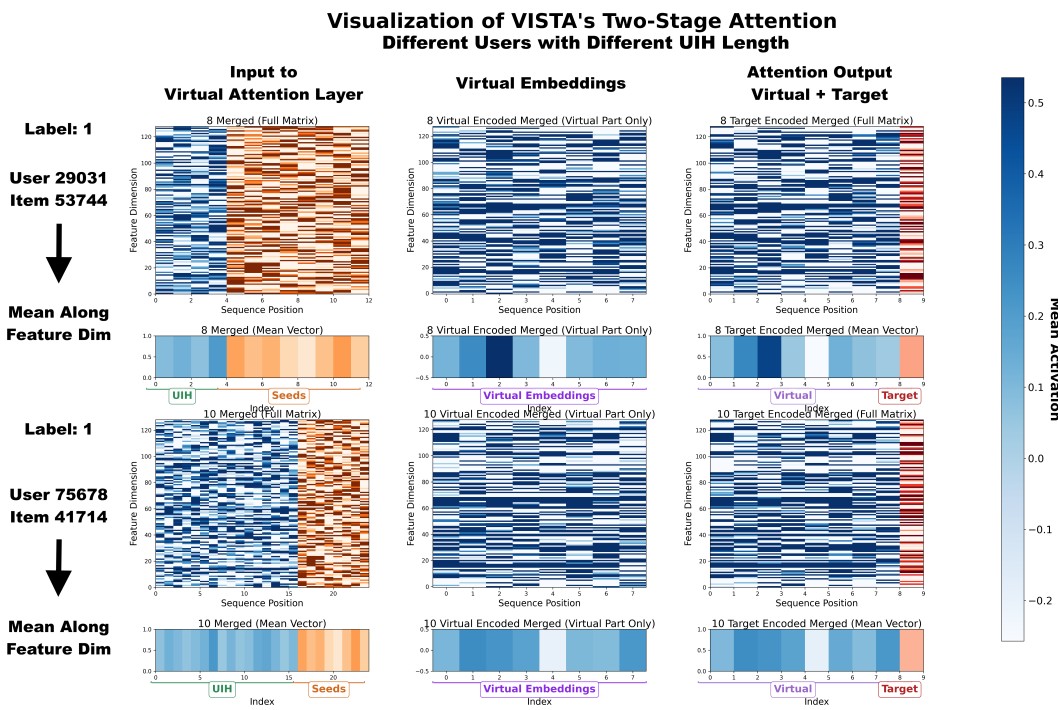

Figure 15: Visualizing the virtual attention and target attention layers for the different users having different UIH sequence lengths (both positive candidates).

