# OpenReview forum: "Massive Memorization with Hundreds of Trillions of Parameters for Sequential Transducer Generative Recommenders"
_ICLR.cc/2026/Conference — ICLR 2026 Poster_

### Official Review · Reviewer_Difm · 2025-10-31

**Soundness:** 3
**Presentation:** 4
**Contribution:** 3
**Rating:** 6
**Confidence:** 4

**Summary:**

The paper introduces VISTA, a two-stage framework for scaling sequential recommendation to lifelong user histories with fixed training and inference costs. Stage 1 compresses ultra-long histories into ~100 cached summary embeddings via quasi-linear attention with virtual seed tokens. Stage 2 lets candidate items attend only to these summaries, avoiding full-history access.
Deployed at billion-user scale, VISTA delivers +2.1% CTR and +1.8% session duration gains with O(1) inference cost and manageable (O(100 TB–1 PB)) storage.

**Strengths:**

1. Summarizes histories once, caches ~100 embeddings, achieving constant-time inference with practical TB–PB storage trade-offs.
2. QLA ensures no label leakage, cleanly separates history and target attention, and yields interpretable user clusters.
3. Uses reconstruction loss for rich summaries and integrates quantization, KV caching, and async updates for production readiness.

**Weaknesses:**

1. Heavy reliance on summary quality and update frequency The effectiveness of VISTA critically depends on the fidelity of Stage 1 summarization embeddings, yet the paper does not specify how frequently these summaries are refreshed in production.
2. Shared virtual seeds limit personalization; smaller summary budgets unexplored The virtual seed tokens used to initialize summarization are shared across all users, which may fail to capture highly idiosyncratic or niche preferences. While PCA visualization shows country-level clustering, it does not prove fine-grained personalization.
3. All candidates attend the same fixed ~100 summary tokens — no dynamic selection or weighting Every candidate item attends to the identical set of precomputed summary embeddings, with no mechanism to dynamically select or reweight subsets based on candidate type or context.

**Questions:**

see weakness

---

> ### Author Response · Authors · 2025-11-21
>
> We thank the reviewer for taking the time to give constructive feedback on our work while recognizing its merits. In the following, we will take care to respond to the points of weakness raised in your review.
>
> * As in our response to Reviewer Jfea, the summarization embedding update process is as follows: New user requests initiate both the training and computation of summarization embeddings. These embeddings are then deduplicated by user ID, with only the most recent embedding retained every two hours and written to a table. A KV store index monitors this table and incrementally indexes new partitions as they appear. We selected a two-hour update cadence because it proved to be the most reliable end-to-end on our systems. Additionally, we experimented with real-time embedding updates by feeding the KV store directly from a streaming source. The results of that A/B test showed a neutral impact. Our hypothesis is that VISTA primarily captures long-term user interests, so very recent engagements (< 2 hours) contribute limited incremental signals to the existing internal recommendation system,  which already anchors on recent user activity. While we had originally focused on the other model designs in the original manuscript within the page limits, we see that this embedding update cadence information will be valuable to add and have **taken care to include this now in the updated version of the paper**. Thank you for your question!
>
> * Regarding shared virtual seeds, we also explored personalizing the seeds by incorporating the User ID embeddings into the seed embedding generation. However, this approach resulted in a 0.11% regression in NE on the primary 'C-Task,' alongside declines in other metrics. We hypothesize that the summarization module already captures sufficient personalization through the attention to the UIH tokens. Intuitively, user profile features—such as country, city, and language—can be easily inferred from a sufficiently long user interaction history. Consequently, the model is likely able to implicitly capture these personalization attributes directly from the behavior data. (as discussed further below). Moreover, we want to clarify that even though all users share the same seed embedding parameters (as inputs), but after the summarization module (which performs self-attention on the user interaction history + shared seeds), the output summarization embeddings are different for each user. (We can see this in, e.g., the Appendix section visualizations.) These summarization embeddings are then used for the target attention.
>
> * Regarding the dynamic selection of summarization tokens, we completely agree that doing such is a great idea and we have considered this route. Specifically, scaling beyond ~100 tokens to thousands would allow for dynamic selection mechanisms (e.g., top-k), which is very likely to improve the model performance. However, this presents a massive infrastructure challenge: storing thousands of tokens for billions of users would require storage on the scale of tens of petabytes (taking replication into account). Thus, we have to defer this feature to future work, after first adopting embedding compression/quantization to solve the storage constraints. Nonetheless, we have been successful at utilizing the current version which generates ~100 tokens for each user in our production systems.

---

> > ### Comment · Reviewer_Difm · 2025-11-25
> > **Reply to authors**
> >
> > Dear authors:
> >
> > Thank you for the clear and well-organized rebuttal. The additional experiments and clarifications directly address my earlier concerns regarding the reliance on summary quality and the limited degree of personalization. The revised manuscript is now stronger and more complete.
> > I am satisfied with the authors’ response and will maintain my score.

---

### Official Review · Reviewer_Jfea · 2025-11-01

**Soundness:** 3
**Presentation:** 3
**Contribution:** 3
**Rating:** 6
**Confidence:** 3

**Summary:**

The paper presents a two-stage framework for large-scale recommendation systems aimed at modeling extremely long user interaction histories while maintaining computational efficiency. The method separates historical summarization from candidate evaluation: user histories are first compressed offline into a fixed number of embeddings, which are then reused online through candidate-specific attention. The authors also introduce a quasi-linear attention variant and a reconstruction-based training loss. Experiments include both public and industrial datasets, with partial deployment results reported.

**Strengths:**

The paper tackles a practically significant challenge in large-scale recommender systems: how to model ultra-long user behavior sequences without overwhelming computational resources efficiently. The idea of decoupling offline summarization and online attention is pragmatic and fits well within production serving architectures. The framework appears to strike a reasonable balance between modeling capacity and latency, and its design aligns with industrial constraints. The experiments cover multiple datasets and include an online A/B test, providing some evidence of real-world feasibility. The overall presentation is clear, with logical flow and illustrative figures that help readers follow the motivation and method.

**Weaknesses:**

W1: The paper presents various performance comparisons, but fails to provide any statistical significance tests for these results, unclear if the differences are due to random seeds; supplementary tests (e.g., a marginal +0.002 AUC gain in Table 1) and related metrics are needed to validate the gains.

W2: While the paper reports metrics like CTR improvements from online A/B tests, the authors did not mention how computational efficacy is in this scenario, which is essential for assessing the real-world applicability of the framework.

W3: Given dynamic user log updates in a real-time scenario, it is unclear whether embedding updates are triggered periodically or by a long enough sequence length. Additionally, there is no clarification on whether updates in on the scale of all parameters and weights or incrementally.

**Questions:**

Q1: For your results compared to baselines, like the marginal AUC gains in Table 1 are only 0.002, very limited, have you conducted statistical significance tests? Could you provide p-values, or variance, or just let me know the reliability of the reported improvements is not due to the random seed?

Q2: About the online A/B tests, could you all talk about computational efficiency performance compared to baseline models? And is the comparison the same in the offline testing?

Q3: Regarding summarization embedding updates, the paper provides vague descriptions of this process, which is critical for handling dynamic user behavior. Is the update triggered by a fixed schedule (e.g., daily) or by the growth of the interaction sequence? Do you adopt full recomputation or incremental learning for updates?

Q4: In comparison to DGIN [1]: Are VISTA and DGIN comparable given their shared focus on lifelong behavior modeling? Could you compare them in terms of methodology, performance, or computational efficiency? (Or at least compare the part of lifelong or a long period of time of behavior sequences modeling.)

[1] Liu, Q., Hou, X., Jin, H., Wang, Z., Lian, D., Qu, T., ... & Lei, J. (2023). Deep Group Interest Modeling of Full Lifelong User Behaviors for CTR Prediction. arXiv preprint arXiv:2311.10764.

---

> ### Author Response · Authors · 2025-11-21
>
> We thank the reviewer for the helpful comments and suggestions while recognizing the merits of our work. We will respond to each of the points raised and the corresponding questions in the following.
>
> * Regarding statistical significance tests, we had opted to report the mean and standard deviation across seeds as is common. Nonetheless, the metrics we report are significant at the p < 0.001 level when using a paired t-test of VISTA against the next best models (using all outputs across all seeds). When there are multiple candidates which are the next best model, we also saw that ANOVA tests are significant at the p < 0.001 level. Despite the relatively modest improvement in metrics, seeing statistically significant differences may suggest that model outputs have very little variance on the dataset across lots of examples (union across all seeds). **This is now mentioned in the updated version.**
>
> * For computational efficiency, we would like to highlight that VISTA is a foundational model paradigm that scales with high efficiency, especially when compared to other non-cachable models tested in this paper. The key advantage is that summarization embeddings only need to be computed once per user and can then be cached for use across multiple downstream applications. This approach eliminates redundant computations during both training and inference. For example, in our online A/B test, VISTA demonstrated **about 94% reduction in inference GPU resource** (measured by inference QPS, Queries Per Second) usage by caching and serving embeddings, rather than re-computing them for every new user request. This efficiency is critical for real-world deployment, as it enables scalable and cost-effective serving of recommendations without sacrificing performance.
>
> * Regarding the embedding updates: New user requests initiate both the training and computation of summarization embeddings. These embeddings are then deduplicated by user ID, with only the most recent embedding retained every two hours and written to a table. A KV store index monitors this table and incrementally indexes new partitions as they appear. We selected a two-hour update cadence because it proved to be the most reliable end-to-end on our systems. Additionally, we experimented with real-time embedding updates by feeding the KV store directly from a streaming source. The results of that A/B test showed a neutral impact. Our hypothesis is that VISTA primarily captures long-term user interests, so very recent engagements (< 2 hours) contribute limited incremental signals to the existing internal recommendation system,  which already anchors on recent user activity. While we had originally focused on the other model designs in the original manuscript within the page limits, we see that this embedding update cadence information will be valuable to add and have **taken care to include this in the updated version of the paper**. Thank you for your question!
>
> Answer to Questions
>
> * As the response above (the first bullet)
>
> * Please see responses above for the computational efficiency in online A/B tests. In the offline setting, we do not cache and retrieve embeddings like in the online setting, thus the inference QPS savings do not apply for offline experiments. However, as mentioned before, we also compared real-time embedding updates and the 2-hour cadence updates in online experiments, which showed neutral impact.
>
> * As the response above (last bullet)
>
> * Thank you for bringing this work to our attention; **we have included it now in our related work**. To answer the question, it appears that DGIN and VISTA fundamentally differ in how long user histories are compressed. In DGIN, it appears that an offline GM module groups behavior into interest groups while keeping statistical and aggregated attributes of the individual items within groups. (This grouping is illustrated in their Figure 3.) It appears that then, the target attention is between the candidate item (target) to the groups (eq. (7)). In addition, there’s a target module which also extracts sub-sequence information from the user history pertaining to the target item. On the other hand, we perform causal self-attention in a hierarchical manner on the full user behavior history (no grouping) concatenated with the shared seed embeddings. The output of this attention layer corresponding to the seed embedding indices are the summarization embeddings, and target attention is done using these virtual embeddings and the candidates. Regarding computational costs, it is likely that our method has higher computational cost during the training stage (since we apply self-attention on the full UIH). Our target attention during inference is likely to be comparable to DGIN since we apply target attention using ~100 tokens, and DGIN has two Interest_GM and Interest_TM calculations; the comparison of which is more costly depends on the number of groups and the length of the target-aware subsequence in DGIN.

---

### Official Review · Reviewer_khLg · 2025-11-01

**Soundness:** 3
**Presentation:** 2
**Contribution:** 2
**Rating:** 4
**Confidence:** 3

**Summary:**

This paper proposed VISTA (VIrtual Sequential Target Attention)， addressing the scalability challenge of processing ultra-long user interaction histories (up to 1M items) in industrial recommendation systems, where traditional Transformer-based models (e.g., HSTU, SASRec) suffer from prohibitive latency and GPU costs. It proposes a two-stage framework: (1) UIH Summarization compresses ultra-long UIH into hundreds of virtual seed embeddings; (2) Target-Aware Attentioncomputes attention between candidate items and cached summary embeddings (bypassing full UIH processing).
The summary embeddings are stored in a distributed key-value system, enabling fixed inference costs regardless of UIH length. Experiments on public (Amazon-Electronics, KuaiRand-1K) and industrial datasets show VISTA outperforms baselines (HSTU, DIN, SASRec)

**Strengths:**

- Interesting design. VISTA decouples UIH processing into offline summarization and online attention. This design enables handling UIH up to 1M items while keeping inference costs fixed—critical for industrial systems serving billions of users with lifelong interaction histories.

- QLA resolves linear attention’s limited expressiveness by integrating SiLU non-linearity and self-target attention.

- Practical Industrial Deployment Design: VISTA includes a distributed embedding delivery system that handles petabytes of summary embeddings. Online A/B tests on a production platform confirm tangible gains with minimal latency, validating its real-world applicability.

**Weaknesses:**

- Reliance on Seed Embedding Quality: The summarization stage’s performance hinges on virtual seed embeddings—experiments show increasing seeds from 64 to 128 improves NE by 0.04–0.12% but raises storage costs exponentially

- Limited Analysis of Reconstruction Loss: The generative reconstruction loss is claimed to enhance information retention, but its contribution is weakly validated. Ablation shows the loss has marginal value.

- Brittleness in Short UIH Scenarios: On public datasets with short UIH (e.g., Amazon-Electronics), VISTA’s performance is not superior.

**Questions:**

What is the information loss of summarization? The paper claims the reconstruction loss retains UIH information, but no quantitative analysis (e.g., mutual information between summary embeddings and full UIH, or other way) is provided

---

> ### Author Response · Authors · 2025-11-21
>
> We thank the reviewer for taking the time to review our work and contribute to the improvement of our manuscript. We will do our best to respond to questions and points of weakness below. We hope that our response and updates to the manuscript will address the points raised.
>
> * Regarding the comment on seed embeddings and storage costs, we would like to clarify that actually the storage should only increase **linearly** with the number of seed embeddings. For every user, increasing the number of seeds from 64 to 128 increases the storage cost from 64 to 128 for each user accordingly. However, we acknowledge that in massive industrial-scale systems, even linear growth can eventually become prohibitive. Thus it is a fundamental tradeoff on deciding the number of seed embeddings considering the (linear) increase in storage cost and increased computational costs during training, and the downstream performance improvement.
>
> * In our ablation study comparing VISTA and VISTA-w/o-Recon (VISTA without reconstruction loss), we show that including the reconstruction loss improves NE for the C-Task by 0.05%, E1-Task by 0.1% and E3-Task by 0.03%. On our platform, the C-Task and E1-Task are usually most important, and these relative improvements are considered significant at this scale of data. We have also taken care to provide additional results exploring the reconstruction loss and summarization in our response to your question (see below).
>
> * Indeed, VISTA was designed to specifically address the unique scalability issues in processing the ultra-long sequences present in proprietary industrial data, which is a few orders of magnitude bigger than the public dataset. Nonetheless, we found it interesting that the model could also be used in shorter sequence settings, to demonstrate its effectiveness. As the virtual seed embeddings can be interpreted as an alternate embedding representation which is influenced by both the individual user history and the globally shared seed parameters, we interpret the results as showing a (mild) improvement in model generalization due to the influence of shared global information. Whether global information improves model performance may also be due to properties of the dataset as well.
> Moreover, we would like to emphasize that modeling longer sequences for recommendation aligns with current industry trends, driven by the widely observed 'scaling law'—where longer user histories almost guarantee improved model performance. While we utilized public datasets primarily to explore its use in other settings, we acknowledge that the results on these smaller benchmarks do not fully reflect the capabilities of the proposed VISTA method.
>
>
> Q: On Reconstruction Loss and Information Loss.
>
> The reconstruction loss itself is a measure of how much information of the full UIH is captured by VISTA’s virtual embeddings. In practice, we used L2 norm to measure how much we can reconstruct the original UIH given the virtual embedding as input, and we have verified that it is an informative metric for us to quantitatively measure the reconstruction quality.
>
> On the Amazon dataset, for example, we can see that training the VISTA model without explicitly minimizing the reconstruction loss still reduces the reconstruction loss of the learned embedding against the full UIH as the model improves. However, the reconstruction loss plateaus and the model takes longer to converge after some training steps (as our training pipeline supports early stopping). With the explicit introduction of the reconstruction loss, we see a dramatic decrease in the reconstruction loss in the first training steps and faster model convergence. The test metrics also improved by 0.22% AUC and 1.11% NE with the use of the reconstruction loss.
>
> Thanks to reviewer KhLg for pointing this out. **We have added this quantitative analysis in Appendix C** for a broader audience who are also interested in this technique.

---

> > ### Comment · Reviewer_khLg · 2025-11-27
> >
> > The rebuttal adequately addresses my concerns. Therefore, I raise my score.

---

### Official Review · Reviewer_osbe · 2025-11-02

**Soundness:** 4
**Presentation:** 4
**Contribution:** 4
**Rating:** 8
**Confidence:** 4

**Summary:**

This paper introduces VIrtual Sequential Target Attention (VISTA), a two-stage approach that solves the critical computational bottleneck in modern sequential recommendation systems. It strikes the balance between computational efficiency and predicitve accuracy, as well as the latency and scalability challenges. The core contributions are the ultra-long UIH summarization and the quasi-linear attention for recommendation, which are well motivated for recommendation scenerios. It has comprehensive experiments with industrial-scale dataset results and online A/B experimental results.

**Strengths:**

1. This paper is well motivated. The scalabity and efficiency are important for industrial recommendation systems. The core two-stage architecture (UIH summarization and target attention) effectively decouptions the quadratic computation of attention from real-time inference.
2. The proposed quasi-linear attention (QLA) with linear complexity is tailored for recommendation.
3. The results are backed by comprehensive offline and online A/B tests on a massive industrial-scale dataset, showing significant real-world improvements.

**Weaknesses:**

On public datasets like Amazon and KuaiRand, the performance gains are marginal compared to baselines. This suggests the primary advantage of VISTA is strictly in the extreme-scale, ultra-long sequence regime of proprietary industrial data, limiting generalizability.

**Questions:**

Although the proposed quasi linear attention is designed for recommendation, there are some other linear/efficient attention architectures for sequence modeling and langauge modeling. Are those existing linear/efficient attention architectures (e.g., Deepseek's FlashMLA) with hardware optimization also effective and efficient for sequential recommendation? If so, they could potentially be used to build recommendation systems for further gains.

---

> ### Author Response · Authors · 2025-11-21
>
> We thank the reviewer for recognizing the merits of our work and appreciate the constructive feedback to improve it. Indeed, VISTA was designed to specifically address the unique scalability issues in processing the ultra-long sequences present in proprietary industrial data, which is a few orders of magnitude bigger than the public dataset. Nonetheless, we found it interesting that the model could also be used in shorter sequence settings, to demonstrate its effectiveness. As the virtual seed embeddings can be interpreted as an alternate embedding representation which is influenced by both the individual user history and the globally shared seed parameters, we interpret the results as showing a (mild) improvement in model generalization due to the influence of shared global information. Whether global information improves model performance may also be due to properties of the dataset as well.
>
> Moreover, we would like to emphasize that modeling longer sequences for recommendation aligns with current industry trends, driven by the widely observed 'scaling law'—where longer user histories almost guarantee improved model performance. While we utilized public datasets primarily to explore its use in other settings, we acknowledge that the results on these smaller benchmarks do not fully reflect the capabilities of the proposed VISTA method.
>
> Regarding the quasi-linear attention, we have continued to iterate on various attention mechanisms since our initial submission, driven by the field's rapid progress. We have experimented with approaches such as FlashMLA, Kimi Linear [1] and even Flash Attention 4 on B200 GPUs, identifying several improvements for subsequent model versions. However, the results reported in this manuscript reflect the production model at the time of writing in July 2025, utilizing the described quasi-linear attention. We intend to publish our findings on these advanced, practically proven mechanisms in future work (targeted for early 2026). We thank the reviewer again for their positive assessment.
>
> [1] Kimi Team et.al. Kimi Linear: An Expressive, Efficient Attention Architecture, November 2025.

---

> > ### Comment · Reviewer_osbe · 2025-11-27
> >
> > Dear authors,
> >
> > Thank you for addressing my concern. I will maintain my positive score for this work.

---

### Meta-Review · Area_Chair_nNwb · 2026-01-11

**Summary:**

This paper proposes VISTA, a two-stage attention framework that decomposes target attention into offline user interaction history (UIH) summarization and online candidate-item attention, aiming to address scalability challenges of ultra-long sequences in industrial recommendation systems. VISTA achieves fixed training/inference costs for lifelong UIHs (up to 1M items) and has been successfully deployed on a billion-user platform with positive offline and online metrics.

The reviewers' key concerns lie in: limited improvements on short sequence data, unclear summarization embedding update cadence, and lack of statistical significance tests for results

**Reviewer Concerns:**

Addressed Concerns:
- Unclear update cadence: Clarified 2-hour update cycle and verified its effectiveness via online A/B tests.
- Lack of statistical tests: Supplemented paired t-test/ANOVA results (p<0.001) to confirm significant improvements.
- Insufficient reconstruction loss analysis: Added quantitative analysis.


Outstanding Concerns:
- Limited improvements on short sequence data remain unaddressed; the model’s advantage is restricted to long industrial sequences.

**Reviewer Scores:**

- Reviewer osbe: Original score 8 (accept), maintains positive score after rebuttal.
- Reviewer khLg: Original score 4 (marginally below acceptance), likely to raise score as rebuttal adequately addresses all concerns.
- Reviewer Jfea: Original score 6 (marginally above acceptance), likely to maintain or slightly increase score as authors addressed  their points.
- Reviewer Difm: Original score 6 (marginally above acceptance), likely to maintain score as rebuttal satisfactorily addresses concerns.

---

### Decision · Program_Chairs · 2026-01-26

Accept (Poster)